# The role of psychological flexibility in the meaning-reconstruction process in cancer: The intensive longitudinal study protocol

**Aleksandra Kroemeke**[1]*, **Joanna Dudek**[2], **Małgorzata Sobczyk-Kruszelnicka**[3]

**1** Faculty of Psychology in Warsaw, Department of Health Psychology, SWPS University of Social Sciences and Humanities, Warsaw, Poland, **2** Faculty of Psychology in Warsaw, Department of Behavioral Psychology, SWPS University of Social Sciences and Humanities, Warsaw, Poland, **3** Department of Bone Marrow Transplantation and Oncohematology, Maria Sklodowska-Curie National Research Institute of Oncology (MSCNRIO), Gliwice, Poland

* akroemeke@swps.edu.pl

**Data Availability Statement:** No datasets were generated or analysed during the current study. Data that will be collected cannot be shared publicly due to ethical reasons (the possibility of

## Abstract

### Objectives

Meaning-making is an important element of adapting to disease. However, this process is still poorly understood and the theoretical model has not been comprehensively verified yet, particularly in terms of complexity, dynamics, and intraindividual variability. The aim of this study is a deeper understanding of the meaning-reconstruction process in cancer and empirical verification of the integrative meaning-making model of coping extended by the psychological flexibility model. We postulate that psychological flexibility can foster the meaning-making in cancer by building more flexible and workable meaning-making explanations of disease.

### Design

A daily-diary study conducted for 14 days in patients following the first autologous or allogeneic hematopoietic cell transplantation (HCT).

### Methods

Participants (at least 150) will be requested to complete the daily-diary related to daily situational meaning, meaning-related distress, meaning-making, psychological flexibility, meanings made, and wellbeing for 14 days after hospital discharge following HCT. Also, baseline and follow-up assessment of global meaning, wellbeing, and meanings made will be performed. Statistical analysis of the data will be conducted using the multilevel and dynamic structural equation modeling.

### Conclusions

The study will fill in the gaps in health psychology in the understanding of the meaning-reconstruction process in cancer by within- and between-person verification of the integrative meaning-making model and its extension by the psychological flexibility model. The

identification of participants). Data will be made available from the Ethical Review Board at SWPS University of Social Sciences and Humanities, Faculty of Psychology in Warsaw (psychoetyka@swps.edu.pl) for researchers who meet the criteria for access to confidential data.

**Funding:** The work is supported by the National Science Centre, Poland [grant number 2020/39/B/HS6/01927 awarded to AK]. The funders had no role in study design, data collection and analysis, decision to publish, or preparation of the manuscript.

**Competing interests:** The authors have declared that no competing interests exist.

data obtained will be used in further research on the development of meaning-making by means of interventions based on psychological flexibility.

## Introduction

Meaning plays a central role in our wellbeing, particularly in high stress conditions such as cancer. An important element of adapting to cancer is meaning reconstruction which is described by the integrative meaning-making model of coping [1]. However, meaning-making to disease is still poorly understood and its theoretical model has not been comprehensively verified yet. Such verification would consider its complexity, dynamics, reciprocity, and intraindividual variability. Therefore, we plan to conduct the intensive longitudinal research aimed at an in-depth understanding of the meaning-reconstruction process in cancer. The theoretical basis of the research will be the integrative meaning-making model of coping [1] extended by the psychological flexibility model [2].

### Integrative meaning-making model of coping

Based on the integrative meaning-making model [1], meaning-making (i.e. process of searching for meaning and explanation for adversity) depends on global meaning (i.e. core schemas through which people perceive themselves and their surroundings, interpret the past, anticipate the future and follow their behavior), situational meaning (i.e. apprised meaning of a particular situation), and distress related to the discrepancy between them. Meaning-making impacts the meanings made (i.e. the perception of positive changes resulting from successful coping with adversity; benefit finding) and then wellbeing, which makes it a mediator in the model [1]. Meaning-making refers to approach-oriented intrapsychic efforts, which involve increasing the matching (or reduction of discrepancies) between global and situational meaning by changing either the meaning of the situation itself or one's global beliefs and goals [1].

The complete verification of the model has not been made. Reports on the relationships between global meaning (i.e. beliefs, goals, sense of purpose) and wellbeing are predominant and show their positive relationships [3], also in the affected persons [4,5]. Situational meaning can play a mediating function in these relationships through more challenging, controllable and less threatening appraisal of an event, albeit so far outside the disease context [1,6]. There are also data indicating that the disease may impair general beliefs and life goals, which is associated with patient higher distress, anxiety and depression and lower quality of life [7–9]. Research on the relationships between meaning-making and adjustment to illness brings inconsistent findings indicating positive [10,11] and negative associations [4,12,13]. The relationships between meaning-making and meanings made are poorly investigated and are inconclusive, although most studies were not disease-related. In cancer survivors, meanings made mediated the association between meaning-making and longitudinal adjustment [9]. However, outside the disease context, meaning-making completed with meanings made was necessary, unnecessary or irrelevant for better adjustment [1]. Most studies focused on the effect of meanings made on adjustment to disease, without a simultaneous assessment of meaning-making. These studies were mostly cross-sectional and indicated positive or insignificant effects [14]. A meta-analysis of benefit finding following various stress conditions found that it was related to less depression and more positive wellbeing, but unrelated to anxiety, global distress, quality of life, or subjective physical health [15]. On the contrary, the review of

longitudinal studies in the disease context found a favorable effect of meanings made on physical aspects of adjustment rather than psychological ones [14].

## Psychological flexibility model

Psychological flexibility is defined as an individual's ability to freely choose the action that is consistent with one's own goals and values, regardless of what thoughts, emotions and impressions accompany it [2]. The psychological flexibility model includes six key processes, i.e. acceptance, cognitive defusion, contact with the present moment, perspective-taking, values clarification, and committed action [2]. Acceptance and defusion are the most important skills that increase a person's openness to direct experiences (through the attitude of interest and curiosity) and allow for diversity in action, which enables people to freely take action consistent with the values of the individual. Contact with the present moment focuses on the experiences of the present and perspective-taking allows one to view them from a broader perspective. Values clarification and committed action bring vitality and meaning to the actions taken through readiness to valuable engaging in life. Due to identification of life values, it is possible to link current behavior with what is important and meaningful. Each key process of psychological flexibility plays a decisive role in a person's ability to adapt to adversity [2].

A number of studies have found that psychological flexibility is associated with more adaptive coping [16], adaptive psychological traits, including higher conscientiousness and openness to experience [17], and better wellbeing, including physical health, quality of life, and emotional wellbeing in the healthy population [2,18]. Favorable results of psychological flexibility on physical and emotional wellbeing were also found in patients with chronic diseases [19,20]. Psychological flexibility has also been proven to be a buffer that mitigates the effects of stress on wellbeing in healthy persons [16] and patients [21]. We have not found any research that would explicitly test the relationships between psychological flexibility and global or situational meaning, meaning-making or meanings made. However, some data indicated that people characterized by openness and curiosity showed a higher tendency to benefit finding [22,23].

## Rationale for the present study

Although the process of meaning-reconstruction in disease is an important element of adaptation, little is known about it. Earlier studies failed to address the complexity or dynamics of the meaning-making process. As a result, the model has not been fully verified and the available data apply only to the between-person differences. The solution to this problem is to conduct the real-life research using the intensive longitudinal approach which allows for studying the dynamic intraindividual variability [24]. Research on the meaning-making process in disease should also consider all the components of the model to determine the adaptive value of the process. Previous research did not differentiate between the process and its results. Most studies did not measure meaning-making and meanings made simultaneously. Therefore, drawing clear conclusions is difficult in terms of whether meaning-making is associated with adjustment to disease to the extent that meaning is made. It is likely that meaning-making adaptability depends on meanings made. However, this has not been clearly confirmed yet [1]. Furthermore, studies should examine moderation and mediation relationships. Data are missing on what mechanisms lie behind the meaning-making process. Completing the model with psychological flexibility seems to be a promising solution. Psychological flexibility promotes acceptance of what is difficult to change or is not subject to change, taking responsibility for one's own experiences and actions, and creating a meaningful life by engaging in activities that are consistent with one's values. Increasing psychological flexibility should therefore foster the

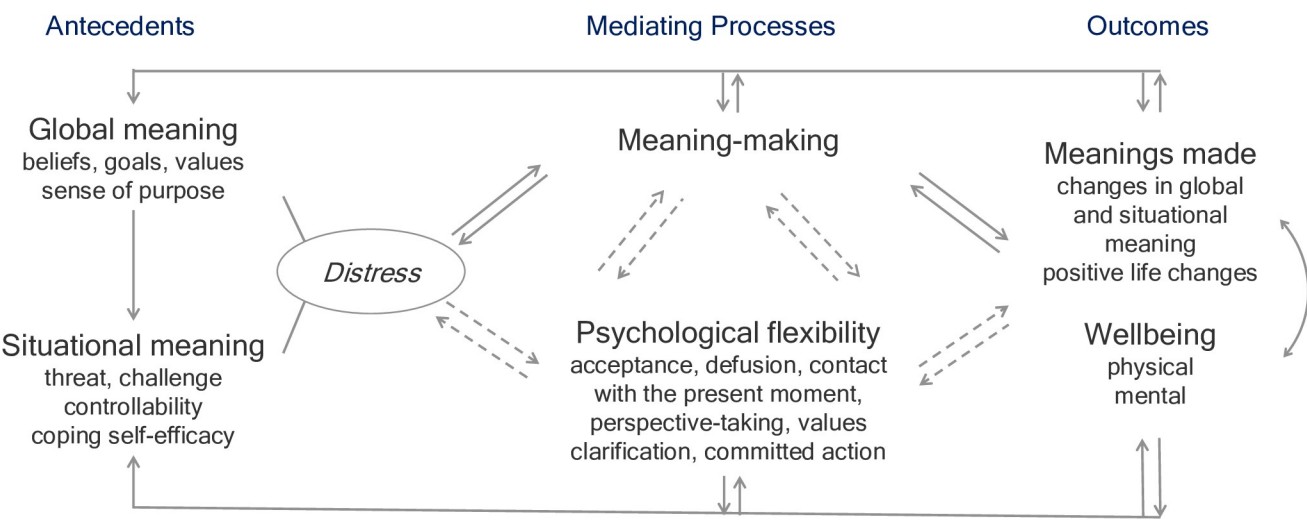

**Fig 1. Research model: The meaning-making model of coping with a chronic disease (solid lines) extended by psychological flexibility (dashed lines).**

creation of meaning in disease by building more flexible and workable meaning-making explanations of disease. Thus, psychological flexibility may constitute the missing mechanism of meaning-making strategies in the integrative meaning-making model.

The purpose of our research will be (*i*) the identification of individual trajectories and sources of variation of the meaning-reconstruction process during a chronic disease, (*ii*) the investigation of psychological flexibility as a possible mediator of the meaning-reconstruction process, and (*iii*) the determination of within-person dynamic and reciprocal associations in the extended meaning-making model (see Fig 1) in an observational intensive longitudinal study among patients following hematopoietic cell transplantation (HCT). HCT is a highly invasive and life-threatening treatment of hematologic neoplasms associated with burdensome adverse effects and a strong medical regimen for patients [25]. However, this procedure gives patients hope for recovery or long-term remission. Therefore, HCT can represent a mini-seismic event during coping with cancer (the turning point in patient lives) posing a challenge to the patient meaning structures. The post-HCT period may prompt reflection on the meaning of HCT and the patient current situation, which are part of meaning-making. In addition, HCT is mostly performed after the period known as the shock and denial phase. Potentially, this is the period during which the patient can start more reflective, meaning-making coping.

## Study research questions and hypotheses

We formulated the following research questions: (*1*) If and what are the individual trajectories of meaning-making across first 14 days after discharge (within-person level) and how do those trajectories differ from person to person (between-person level)? (*2*) What processes underlie the individual meaning-making fluctuations and how do the patients differ in this process? In particular: (*a*) Does daily meaning-making mediate the effect of daily fluctuations in situational meaning and distress on fluctuations in meanings made and wellbeing? (*b*) Does daily meaning-making mediate the effect of baseline to follow-up changes in global meaning, meanings made, and wellbeing? (*c*) Do daily meanings made moderate the association between daily fluctuations in meaning-making and wellbeing? (*d*) Does psychological (in)flexibility mediate the effect of daily fluctuations in distress on fluctuations in meaning-making? (*e*) Does psychological (in)flexibility mediate the effect of daily fluctuations in meaning-making

on fluctuations in meanings made? (*f*) Do these relationships occur when the direction of the relationships between variables is reversed? (*g*) Do demographic, situational and individual factors moderate the individual trajectories and tested relationships?

We expect that various trajectories of meaning-making following HCT will be identified with various directions (increase vs. decrease) and dynamics over time (*Hypothesis 1; H1*). In addition, we expect the support for assumptions of the meaning-making model, importance of daily psychological (in)flexibility in this process, and reciprocal associations between variables in the extended meaning-making model. Based on the integrative meaning-making model, we predict that daily meaning-making will mediate between daily fluctuations in antecedents/distress and outcomes (*H2*), as well as between baseline and follow-up changes in global meaning and outcomes (*H3*). Also, positive effects of daily fluctuations in meaning-making on wellbeing will occur on the days when meaning is given (*H4*). Referring to the psychological flexibility model, we assume that daily psychological flexibility can determine meaning-making and therefore can mediate between daily distress and meaning-making (*H5*). Alternatively, daily psychological flexibility can be determined by meaning-making and can mediate between daily meaning-making and meanings made (*H6*). Without hypotheses, but more exploratively, we also postulate the existence of additional pathways of relationships of a reciprocal nature, both in relation to the relationships between distress, psychological (in)flexibility and meaning-making, and the other elements of the model, i.e. global and situational meaning, as well as outcomes of the meaning-making process. Finally, we suppose that demographic (e.g. age and sex of patients), situational (e.g. time since the diagnosis) and individual factors (e.g. global and situational meaning, baseline wellbeing) could be potential moderators of the meaning-reconstruction process.

## Materials and methods

### Ethics

The study received ethical approval from the Ethical Review Board at SWPS University of Social Sciences and Humanities, Faculty of Psychology in Warsaw (Decision No. 26/2022 of April 12, 2022) and adheres to the ethical guidelines of the Declaration of Helsinki. All participants will be requested to give written informed consent prior to participation.

### Design

This is an observational study with an intensive longitudinal design. Patients admitted for autologous (patient's own stem cells) or allogeneic (donor stem cells) HCT will be recruited.

### Setting

Patients will be recruited from the Department of Bone Marrow Transplantation and Oncohematology of the Maria Sklodowska-Curie National Research Institute of Oncology (MSCNRIO) Gliwice Branch, Poland. Approximately 215 primary transplants are performed there annually (approx. 246 HCT in total).

### Participants and sample size

The sample size was estimated using a target power of 80%, at alpha of 0.05, and was calculated relative to the small effect size in the latent growth analysis using an a-priori sample size calculator [26]. A minimum of 125 patients should be sufficient to obtain the adequate power analyses. Considering the potential attrition rate of 20%, the final sample is 150 patients. Similar

values (for two-level model with random slopes and within-person mediation model) come from simulation studies using the Monte Carlo approach [24,27].

The inclusion criteria will include qualification of the patient for the first autologous or allo-geneic HCT due to hematologic and lymphatic cancer, $\geq$ 18 years, and written informed consent. The exclusion criteria will be as follows: the presence of any other major medical or psychiatric disorder other than cancer that would impede the ability to participate in the study, insufficient reading and writing skills of patients, and the evidence of patient unreliability.

## Recruitment process

Recruitment will start after elective admission to the transplantation unit due to HCT. We assume that it will take place approximately two days after admission and before the conditioning treatment. Every two days, research team member (AK) will review the lists of patients enrolled for HCT. Then, another research team member (recruiter) will ask patients who met the study criteria about their interest in participating in the study. Those interested will attend an individual initial meeting during which the recruiter will describe the course of the study in detail and will ask for consent to participate. The patient will be enrolled in the study if they provide written informed consent. The consent can be withdrawn at any time without any negative consequences for participants. The recruitment is anticipated to cover the period from May 2022 to January 2024 (approx. 21 months; considering the decline rate of approximately 40%), depending on the COVID-19 pandemic. Participants will be paid PLN 100 (~USD 22) for the participation in the study.

## Data collection methods

Data will be collected via self-reported electronic surveys at baseline and follow-up, and a 14-day diary (Table 1). The baseline survey will be administered directly after obtaining written informed consent. Daily-diary will start on the second day after hospital discharge and will take 14 days. A follow-up survey will be administered on Day 15. All tools within the diary procedure will be shortened so that the number of items measuring any given indicator ranges

**Table 1. Standard protocol items of the study.**

| Time points | Enrollment + Baseline | | Day1 | Day2 | (. . .) | Day9 | Day14 | Follow-up |
|---|:---:|---|:---:|:---:|:---:|:---:|:---:|:---:|
| *Enrollment* | | transplant engraftment . . . isolation . . . hospital discharge | | | | | | |
| Identification | X | | | | | | | |
| Eligibility screening | X | | | | | | | |
| Informed consent | X | | | | | | | |
| *Data collection* | | | | | | | | |
| Demographics | X | | | | | | | |
| Clinical data | | | | | | | | X |
| Global meaning | X | | | | | | | X |
| Situational meaning | | | X | X | X | X | X | |
| Distress | | | X | X | X | X | X | |
| Meaning-making | | | X | X | X | X | X | |
| Psychological flexibility | | | X | X | X | X | X | |
| Meanings made | X | | X | X | X | X | X | X |
| Wellbeing | X | | X | X | X | X | X | X |

Day1-Day14 = days of daily-diary study after post-HCT hospitalization discharge; Follow-up = Day 15.

from two to six, which is a common practice in such research [24]. Participants who will give their written approval will receive daily text messages that will remind them to complete a diary. During the study, research team member (JD) will contact the participant by phone to resolve any issues and answer questions.

## Measures

Patients will be asked to complete the following measures at baseline and follow-up:

**Global meaning.** Global meaning will include a sense of meaning and illness perception, which will be assessed using the 10-item Meaning in Life Questionnaire (MLQ) [28] and the 8-item Illness Perception Questionnaire (B-IPQ) [29], respectively.

**Meanings made.** Meanings made (i.e. positive psychological changes) will be measured using the modified 10-item Post-Traumatic Growth Inventory-Short Form (C-PTGI-SF) [30,31]. The modification consists in changing the wording of the items to refer to the current state instead of recalling and comparing pre- and post-event status [31].

**Wellbeing.** Wellbeing will include symptoms of depression, anxiety, loneliness, and the indicators of health-related quality of life (HRQOL), which will be assessed with the 10-item Centre for Epidemiological Studies Depression Scale Short Form (CES-D-SF) [32,33], the 7-item Generalized Anxiety Disorder Scale (GAD-7) [34], the 20-item Revised UCLA Loneliness Scale [35], and the 30-item EORTC QLQ-C30 Questionnaire [36,37], respectively.

In daily-dairy, patients will be asked to complete the following measures:

**Situational meaning.** Daily situational meaning will be measured using indicators of primary and secondary appraisal of the current situation following HCT using five items from the Stress Appraisal Measure (SAM) adapted to the daily procedure and study context [38]. The secondary appraisal will be additionally measured with six items from the Coping Self-Efficacy Scale (CSE) adapted to this study [39].

**Distress.** Daily meaning-related distress refers to daily beliefs and goal violation and will be assessed using selected nine items from the Global Meaning Violation Scale (GMVS) adapted to the daily procedure and study context [40].

**Meaning-making.** Daily meaning-making will be measured using selected six items from the Core Beliefs Inventory (CBI) [41] and eight items from the Perceived Ability to Cope with Trauma (PACT) scale adapted to the daily procedure [42]. Both tools measure meaning-making efforts, i.e. CBI—reconsideration of global beliefs [40], whereas PACT—remembering the event and reflecting on its meaning (subscale: trauma focus) [43].

**Meanings made and wellbeing.** Daily meanings made will be measured using five items from the modified Post-Traumatic Growth Inventory-Short Form (C-PTGI-SF) [31]. Daily wellbeing will include daily somatic symptoms and affect, which will be assessed using the self-reported symptom checklist [44] and the 12-item adjective scale (reflecting positive and negative affect of various arousal) based on the circumplex model of emotion by Larsen and Diener [45], respectively.

**Psychological flexibility.** Psychological flexibility will be measured with the short form of the Multidimensional Psychological Flexibility Inventory (MPFI) adopted to the daily procedure. This inventory assesses six flexible and six inflexible processes [46].

**Other measures.** At baseline, the demographic data (i.e. age, gender, education, marital status, employment, having children, socioeconomic status) will be assessed. Clinical data (i.e. diagnosis, time since diagnosis, type of HCT, conditioning, concomitant diseases, treatment toxicity, complications after HCT e.g. graft-versus-host disease in allogeneic HCT recipients) will be obtained from the medical records by a physician (MSK).

In the original and Polish versions, all the tools (except GMVS whose psychometric properties will be tested in this study) are characterized by satisfactory psychometric properties. Table 1 shows the standard protocol items of the study.

## Statistical analysis

Analyses will be conducted using the latest Mplus statistical package [47] and IBM SPSS (IBM Corp.; Armonk, NY). We will use the standard $p < .05$ or 95% confidence interval for determination of value probability. The collected data will be first analyzed in terms of sample characteristics and comparisons (frequency, descriptive statistics; ANOVA, t-test or their nonparametric counterparts; $Chi^2$; Pearson's or Kendall's correlation), missing data (frequency, multilevel modeling), and sample attrition (logistic regression analysis). Multilevel confirmatory factor analysis (MCFA) will be performed to calculate the indicator reliabilities (omega coefficient) at the within- and between-person levels and establish the respective measurement models [24,48]. To examine research questions, the multilevel (MSEM) and dynamic structural equation modeling (DSEM) will be applied [24,49]. Both methods allow for the examination of time course, simple between- and within-person associations, and more advanced associations such as mediations and moderations. Moreover, they allow the most recent flexible approach to the missing data (the full information maximum likelihood) [50,51], which is possible due to the setting and daily-diary procedure.

Hypotheses 1–6 and the research question 2g will be mostly verified using MSEM. In MSEM, random coefficient models with maximum likelihood as an estimator will be applied. Predictors will be divided into within-person (the deviation from the person mean) and between-person (stable between-person mean for each person across all their diary days) indicators, which allowed separation of within-person change from between-person differences of the predictor. Centered linear time trend will be controlled in the analyses. In all models, possible confounders (i.e. demographics, clinical factors, other confounders) will be considered after preliminary selection. For exploratory reasons, lagged models that predict outcomes based on the previous-day predictors will also be considered. Significant interactions (H4, question 2g) will be graphed and probed with simple slope analyses [52].

Research question 2f and H3 will be verified using a multilevel VAR(1) model in DSEM (other hypotheses and questions can also be verified with DSEM). VAR(1) model consists of a set of regression equations, in which each endogenous variable is regressed on its own lagged values (autoregression) and the lagged values of the other variables (cross-regression) for each individual, which allows for the estimation of reciprocal associations. We will use the Mplus default priors (mean = 0, variance = $10^{10}$) and the Bayesian estimator (specific to DSEM). Within- and between-person associations will be automatically distinguished in DSEM. To compare the strength of cross-lagged associations, we will use the within standardization (i.e. standardization using the within-person variance) [53].

## Limitations of the study

A limitation of the research is the restriction of observation to post-HCT patients undergoing the first transplant engraftment. However, this subgroup allows for gender balance compared to other frequently observed subgroups (e.g., breast cancer women or prostate cancer men), and for the observation of the meaning-reconstruction process after the challenging treatment procedure. Furthermore, the assumed observation time (i.e. 14 days), may turn out to be insufficient to observe the effects of fluctuations in the meaning-making process (if these processes fluctuate at longer intervals than overnight or effects of their fluctuations are time-postponed).

In that case, however, the research will provide valuable information on the dynamics of meaning-making processes.

## Conclusion

The presented research project will fill in the gaps in health psychology in the understanding of the meaning-reconstruction process in cancer and the mechanisms of this process by a comprehensive verification of the integrative meaning-making model and its extension by the psychological flexibility model. The data obtained in the study will be used to design the experimental research on the effects of the psychological flexibility-based intervention on the meaning-making process.

## Acknowledgments

We thank Prof. Sebastian Giebel (head of the Department of Bone Marrow Transplantation and Oncohematology in Maria Sklodowska-Curie National Research Institute of Oncology (MSCNRIO) Gliwice Branch) for agreeing to conduct research in the clinic.

## Author Contributions

**Conceptualization:** Aleksandra Kroemeke.

**Data curation:** Aleksandra Kroemeke.

**Funding acquisition:** Aleksandra Kroemeke.

**Methodology:** Aleksandra Kroemeke, Joanna Dudek.

**Project administration:** Aleksandra Kroemeke.

**Resources:** Aleksandra Kroemeke, Joanna Dudek, Małgorzata Sobczyk-Kruszelnicka.

**Software:** Aleksandra Kroemeke.

**Supervision:** Aleksandra Kroemeke.

**Validation:** Aleksandra Kroemeke.

**Visualization:** Aleksandra Kroemeke.

**Writing – original draft:** Aleksandra Kroemeke, Joanna Dudek, Małgorzata Sobczyk-Kruszelnicka.

**Writing – review & editing:** Aleksandra Kroemeke, Joanna Dudek, Małgorzata Sobczyk-Kruszelnicka.

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
