## [Decision Letter · Decision Letter 0]

7 Jun 2022

PONE-D-21-36245

The role of psychological flexibility in the meaning-reconstruction process in cancer: the intensive longitudinal study protocol

PLOS ONE

Dear Dr. Kroemeke,

Thank you for submitting your manuscript to PLOS ONE. After careful consideration, we feel that it has merit but does not fully meet PLOS ONE’s publication criteria as it currently stands. Therefore, we invite you to submit a revised version of the manuscript that addresses the points raised during the review process.

We look forward to receiving your revised manuscript.

Kind regards,

Professor Benjamin Tan, BNSc MMed PhD RN

Journal Requirements:

2. Thank you for stating the following financial disclosure: "The work is supported by the National Science Centre, Poland [grant number 2020/39/B/HS6/01927]."

Reviewers' comments:

Reviewer's Responses to Questions

**Comments to the Author**

1. Does the manuscript provide a valid rationale for the proposed study, with clearly identified and justified research questions?

Reviewer #1: Yes

Reviewer #2: Yes

2. Is the protocol technically sound and planned in a manner that will lead to a meaningful outcome and allow testing the stated hypotheses?

Reviewer #1: Yes

Reviewer #2: Partly

3. Is the methodology feasible and described in sufficient detail to allow the work to be replicable?

Reviewer #1: Yes

Reviewer #2: Yes

4. Have the authors described where all data underlying the findings will be made available when the study is complete?

Reviewer #1: Yes

Reviewer #2: No

5. Is the manuscript presented in an intelligible fashion and written in standard English?

Reviewer #1: Yes

Reviewer #2: Yes

6. Review Comments to the Author

You may also provide optional suggestions and comments to authors that they might find helpful in planning their study.

Reviewer #1: The rationale for the study and the science are good. Here are my comments:

1- The authors might want to include 'evidence of patient unreliability' as an exclusion criterion

2- If possible, the authors should provide some data about how many patients receive an HCT at the institution where the study will be held, and an estimate as to how many patients might fit the inclusion criteria. This will allow the authors to justify their timeline for data gathering.

3-Small detail: the authors mention that the participants will receive PLN 100 for their participation. I assume that the equivalent provided, ~$25, is in American dollars, in which case it should read USD 25.

4- The authors mention that all the questionnaires have good psychometrics in their original version, but do they have versions in the Polish language with proper reliability and validity? If the questionnaires have been translated and the Polish versions have validation data, the authors should mention this. If the questionnaires have not been translated, the authors should explain whether they will have the questionnaires translated professionally, or whether they will provide their own translation. How the questionnaires are translated for a Polish population may affect the quality of the study and may need to be mentioned as a limitation. In any case, the authors should plan to calculate the split-half reliability of their questionnaires (coefficient alpha) for their sample. This will give an indication of the quality of the questionnaire data, and reliability also affects the strength of the correlations. This is important since this is largely a correlational study.

Reviewer #2: The aim of this study is clear; the validation of the (undlying proces of) the Meaning-Making model and extension thereof with the psychological flexibility model.

Design

The design of the study is longitudinal with a basic measure (after inclusion), during 10 days several measurements of different variables during hospitalization and treatment, and final measure op day 11. The longitudinal design is logical, but I have doubts about the short duration. I am not an expert regarding the target population but I think that Meaning-making is:

1) Not exclusive is during hospitalization and treatment or even does not happen during this period because the focus is on treatment and somatic.

2) That these processes (meaning-making and psychological flexibility) hardly fluctuate during 10 to 11 days. Consequently a longer period is needed to notice changes within the person and/or outcome. The authors indicate this point as a limitation.

It is not clear to me why this population was selected. Have these patients a good chance of healing or are there usually permanent limitation which justifies that research into meaning-making is important for this population is ? Alternatively this population may be selected for practical reasons because of the multiple measurement during the isolation. I strongly suggest to specify the reason.

Method

In some cases only subscales from a questionnaire are used and other scales are shortened (justified by ref 24). Can the authors explain how this impact the validity of the measurements?

There may be a problem with the similarities of the proces variables (acceptance, and values out of psychological flexibility measurement) and the outcome variables (acceptance and sense of meaning in life from the psycholocical flexibility measurement). I have the impression that the overlap is so great that you measure approximately the same, what influences the results.

The outcome well being is measured for depression, anxiety, lonelyness and a HRQL measurement that is primarely oriented towards the symptoms. I can imagine that the subject meaning-making formulated as a positively formulated outcome measure of wellbeing could be chosen as supplement.

The power calculation is done for 150 persons. Considering the large number of variables (appoximately 17) this seems rather low. A statistician should be consulted to clarify this.

7. PLOS authors have the option to publish the peer review history of their article (what does this mean?). If published, this will include your full peer review and any attached files.

Reviewer #1: No

Reviewer #2: No

---

## [Author Response · Author response to Decision Letter 0]

25 Jun 2022

Dear Editor and Reviewers,

Re: Manuscript reference No. PONE-D-21-36245

Please find attached a revised version of the manuscript “The role of psychological flexibility in the meaning-reconstruction process in cancer: the intensive longitudinal study protocol” which we would like to re-submit for publication as a study protocol in the Plos ONE.

We wish to thank the Editor and the Reviewers for their precious suggestions and remarks concerning the manuscript. We found all the comments very useful as they provided insightful guidance on how to improve the manuscript’s quality. We highly appreciate the given chance to let us revise our manuscript and re-submit it to Plos ONE.

Please find our point-by-point responses to each comment of the Editor and the Reviewers. We provided two copies of the manuscript, in one of which all the corresponding changes in the manuscript are shown in the red font. We confirm that the manuscript meets Plos ONE’s style requirements. We also added the statement what role the funders will take in the study and the role it took in the preparation of this manuscript (The funders had no role in study design, data collection and analysis, decision to publish, or preparation of the manuscript). Finally, we included captions for Supporting information files (ERB approval and funder confirmation) at the end of the manuscript.

We have also provided additional changes in the manuscript. We decided to move the diary measurement to the post-hospital period and therefore extend it to 14 days. 14 days of intensive follow-up is the standard period for this type of study. In addition, we decided to include in the study also patients qualified for allogeneic HCT. This group of patients is characterized by a longer time than the diagnosis, which will allow us to better test the moderating effect of this variable for the studied dependencies. There are still no unequivocal data on this subject in the literature. The modified study protocol received the approval of the Ethical Review Board.

We have indicated that data from this study will be available upon request. There are ethical restrictions on sharing a de-identified data set. The identification of each participants will be possible based on the demographics and clinical data, hence, the data will contain sensitive information and its publication will be able to infringe the anonymity of the participants. The Ethical Review Board at SWPS University of Social Sciences and Humanities, Faculty of Psychology in Warsaw may provide access upon request (psychoetyka@swps.edu.pl).

We hope that the revisions made in the manuscript and the accompanying responses will now be sufficient to make the manuscript suitable for publication.

We are looking forward to hearing from you at your earliest convenience.

Sincerely,

Authors

Review Comments to the Author

Reviewer #1: The rationale for the study and the science are good. Here are my comments:

Response: We thank the reviewer very much for the overall evaluation of the study and its rationale.

1- The authors might want to include 'evidence of patient unreliability' as an exclusion criterion

Response: We thank the reviewer for this suggestion. We will be checking the data in this regard, so we have taken this suggestion into account. Thank you.

2- If possible, the authors should provide some data about how many patients receive an HCT at the institution where the study will be held, and an estimate as to how many patients might fit the inclusion criteria. This will allow the authors to justify their timeline for data gathering.

Response: We thank the reviewer for this suggestion. Annually, about 215 patients (who meet the criteria of the first autologous or allogeneic HCT) are admitted to the clinic. Among them, there may be people who do not meet the other criteria (the absence of any other major medical or psychiatric disorder other than cancer that would impede the ability to participate in the study and sufficient reading and writing skills), approx. 5%. This gives about 205 patients a year who meet the criteria. We estimate that approx. 40% of them will consent to participate in the study (approx. seven people per month). Hence, we estimate that the study will take about 21 months. It was similar in our earlier study in this group of patients. However, we reckon with the fact that this time, a pandemic situation may extend the time of data collection. We included this information in the Setting and Recruitment process section:

“Patients will be recruited from the Department of Bone Marrow Transplantation and Oncohematology of the Maria Sklodowska-Curie National Research Institute of Oncology (MSCNRIO) Gliwice Branch, Poland. Approximately 215 primary transplants are performed there annually (approx. 246 HCT in total).” (p. 10)

“The recruitment is anticipated to cover the period from May 2022 to January 2024 (approx. 21 months; considering the decline rate of approximately 40%), depending on the COVID-19 pandemic. Participants will be paid PLN 100 (~USD 22) for the participation in the study.” (p.11)

3-Small detail: the authors mention that the participants will receive PLN 100 for their participation. I assume that the equivalent provided, ~$25, is in American dollars, in which case it should read USD 25.

Response: We thank the reviewer for paying attention to it. We corrected the error and adjusted the dollar amount based on the current exchange rate.

4- The authors mention that all the questionnaires have good psychometrics in their original version, but do they have versions in the Polish language with proper reliability and validity? If the questionnaires have been translated and the Polish versions have validation data, the authors should mention this. If the questionnaires have not been translated, the authors should explain whether they will have the questionnaires translated professionally, or whether they will provide their own translation. How the questionnaires are translated for a Polish population may affect the quality of the study and may need to be mentioned as a limitation. In any case, the authors should plan to calculate the split-half reliability of their questionnaires (coefficient alpha) for their sample. This will give an indication of the quality of the questionnaire data, and reliability also affects the strength of the correlations. This is important since this is largely a correlational study.

Response: We thank the reviewer for this comment. We have added the information about good psychometrics in the Polish version language. All tools (except SAM, CSE, and GMVS) have official Polish versions. SAM, CSE, and GMVS have been adopted by us in line with the art (back-translation method). SAM and CSE were already used by us in an earlier project in the same study group. The tools were characterized by very good psychometric properties. We have prepared GMVS for this project. The psychometric properties of this tool will be tested in the study. In a pilot study, we checked the understanding of items by potential study participants. It is not a complicated questionnaire (in part B it consists of the phrases: physical health, intimacy, social support, and community), the participants agreed that the items are clear and understandable, so we do not expect problems with the reliability or factor structure of this tool. 

“In the original and Polish versions, all the tools (except GMVS whose psychometric properties will be tested in this study) are characterized by satisfactory psychometric properties. Table 1 shows the standard protocol items of the study.” (p. 13)

Of course, as in any diary study, all tools are modified for this procedure (change to instructions: asking about beliefs, emotions, behavior on a given day; matching the context: questions in the context of the patient's current health situation). It is a standard procedure in this type of research ensuring its ecological validity.

Regarding the reliability of the tools, we did not write this, but of course, it will be checked before starting the main analyses. We have added the appropriate statement in the description of the planned statistical analysis. Since the data will be hierarchical, reliability will be measured using the omega coefficient (Bolger & Laurenceau, 2013; Shrout & Lane, 2012). Using the Cronbach’s alpha coefficient for diary methods (when the primary analysis is a within-person analysis of change) is criticized (Shrout & Lane, 2012). In Cronbach’s alpha coefficient or classical test theory, all items are equally related to the underlying construct. For this reason, the use of the Cronbach’s alpha coefficient for within-person level (Level-1) data is inappropriate. We will perform the multilevel confirmatory factor analysis (MCFA), which allows for the specification of separate within-person and between-person reliability (and calculation of omega reliability at each level of analyses). 

“Multilevel confirmatory factor analysis (MCFA) will be performed to calculate the indicator reliabilities (omega coefficient) at the within- and between-person levels and establish the respective measurement models [24,48].” (p. 15)

We would also like to inform you about the other modifications made. We decided to move the diary measurement to the post-hospital period and therefore extend it to 14 days. 14 days of the intensive longitudinal study is the standard period for this type of study (Bolger & Laurenceau, 2013; Mehl & Conner, 2012). In addition, we decided to include in the study also patients qualified for allogeneic HCT. This group of patients is characterized by a longer time than the diagnosis, which will allow us to better test the moderating effect of this variable for the studied dependencies. There are still no unequivocal data on this subject in the literature (Ochoa Arnedo et al., 2019; Park, 2011).

Reviewer #2: The aim of this study is clear; the validation of the (undlying proces of) the Meaning-Making model and extension thereof with the psychological flexibility model.

Response: We thank the reviewer very much for the overall evaluation and for summarizing the study aim.

Design

The design of the study is longitudinal with a basic measure (after inclusion), during 10 days several measurements of different variables during hospitalization and treatment, and final measure op day 11. The longitudinal design is logical, but I have doubts about the short duration. I am not an expert regarding the target population but I think that Meaning-making is:

1) Not exclusive is during hospitalization and treatment or even does not happen during this period because the focus is on treatment and somatic.

2) That these processes (meaning-making and psychological flexibility) hardly fluctuate during 10 to 11 days. Consequently a longer period is needed to notice changes within the person and/or outcome. The authors indicate this point as a limitation.

Response: We thank the reviewer for this comment. We decided to move the diary measurement to the post-hospital period. This was mainly due to the organizational reasons of the hospital, beyond our control. But we are sure that this change will make the study more ecologically valid. As suggested by the reviewer, the deliberative process over the meaning of transplantation may become more active in the post-hospital period. This change extended the diary measurement to 14 days. Two weeks is a standard for this type of research (Bolger & Laurenceau, 2013; Mehl & Conner, 2012). Importantly, we are primarily interested in the effects of day-to-day fluctuations in variables, and not changes throughout the study period. We took into account the length of the study in its limitations, but as a rule, the duration of the study has a greater influence on the observed changes in the variables over time, and not on their daily fluctuations. Therefore, the 14-day period seems to be sufficient to observe the effects of interest to us.

In addition, we decided to include in the study also patients qualified for allogeneic HCT. This group of patients is characterized by a longer time than the diagnosis, which will allow us to better test the moderating effect of this variable for the studied dependencies. There are still no unequivocal data on this subject in the literature (Ochoa Arnedo et al., 2019; Park, 2011).

It is not clear to me why this population was selected. Have these patients a good chance of healing or are there usually permanent limitation which justifies that research into meaning-making is important for this population is ? Alternatively this population may be selected for practical reasons because of the multiple measurement during the isolation. I strongly suggest to specify the reason.

Response: We thank the reviewer for this comment. We have modified the rationale for why it is worth measuring the meaning-making process in this group and this period.

“The purpose of our research will be (i) the identification of individual trajectories and sources of variation of the meaning-reconstruction process during a chronic disease, (ii) the investigation of psychological flexibility as a possible mediator of the meaning-reconstruction process, and (iii) the determination of within-person dynamic and reciprocal associations in the extended meaning-making model (see Fig 1) in an observational intensive longitudinal study among patients following autologous hematopoietic cell transplantation (HCT). HCT is a highly invasive and life-threatening treatment of hematologic neoplasms associated with burdensome adverse effects and a strong medical regimen for patients [25]. However, this procedure gives patients hope for recovery or long-term remission. Therefore, HCT can represent a mini-seismic event during coping with cancer (the turning point in patient lives) posing a challenge to the patient meaning structures. The post-HCT period may prompt reflection on the meaning of HCT and the patient current situation, which are part of meaning-making. In addition, HCT is mostly performed after the period known as the shock and denial phase. Potentially, this is the period during which the patient can start more reflective, meaning-making coping.” (pp. 7-8) 

Method

In some cases only subscales from a questionnaire are used and other scales are shortened (justified by ref 24). Can the authors explain how this impact the validity of the measurements?

Response: The decision to choose items have depended on the number of items making up a given subscale. The optimal number of items per one indicator in the diary study is 5 (Bolger & Laurenceau, 2013). To be able to count the reliability of a given indicator, at least 2 items are needed. We created the protocol in such a way that - if possible - we should include at least 5 items for the indicator of a given variable (minimum 2). We managed to do it for almost all daily-diary measures (SAM – 5 items; CSE – 6 items, 2 per subscale; GMVS – 5 and 4 for each subscale; CBI – 6 items; C-PTGI-SF – 5 items; daily affect – 6 items per subscale). For example, it was not possible for psychological flexibility. The questionnaire MPFI-24 consists of 2 items for a given indicator, so we had to use it in its entirety. Besides, we included 8 items of PACT as meaning-making is a crucial variable in our study. Also, somatic symptoms are measured by a much greater number of items than five, because they cover possible somatic ailments of people after transplantation.

There may be a problem with the similarities of the proces variables (acceptance, and values out of psychological flexibility measurement) and the outcome variables (acceptance and sense of meaning in life from the psycholocical flexibility measurement). I have the impression that the overlap is so great that you measure approximately the same, what influences the results.

Response: We thank the reviewer for paying attention to it. We also had such concerns when we started this project. For this reason, we did not select the most commonly used measures for meaning-making such as acceptance and positive reframing from the COPE questionnaire. We were afraid that they would contaminate the measurement of psychological flexibility. In addition, we selected two tools (PACT and CBI) for meaning-making measurement. While the former may interfere with psychological flexibility, the latter should not (the items relate to other aspects of experience processing). For this reason, we also selected 8 PACT items for the study – to be able to remove those that will contaminate the MPFI-24. In addition, we will perform the multilevel confirmatory factor analysis (MCFA) which will not only allow us to estimate the reliability of the indicators but also test the measurement models before starting the specific analyzes (Bolger & Laurenceau, 2013; Shrout & Lane, 2012).

The outcome well being is measured for depression, anxiety, lonelyness and a HRQL measurement that is primarely oriented towards the symptoms. I can imagine that the subject meaning-making formulated as a positively formulated outcome measure of wellbeing could be chosen as supplement.

Response: Thank you for your attention. It may look like this. In the daily measurements of wellbeing, the proportion between pathogenetic and salutogenic indicators is maintained. In baseline and follow-up measurements, quality of life is supposed to be a salutogenic indicator. We have not decided to include, for example, Ryff’s SWB, because it may contaminate the acceptance measures (psychological flexibility) and sense of meaning (global meaning). In addition to well-being, we are going to see if the global sense of meaning (positively oriented) is changing (pre-HCT to post-HCT), which corresponds to what the reviewer wrote about.

The power calculation is done for 150 persons. Considering the large number of variables (appoximately 17) this seems rather low. A statistician should be consulted to clarify this.

Response: We thank the reviewer for this comment. We based the group calculation on statistical calculations. Similar values (125 and 140, respectively) come from the simulation study (for multilevel structural equation modeling: two-level models with random slopes and within-person mediation models) based on the Monte Carlo approach (Bolger & Laurenceau, 2013; Sagan, 2019). We have added this information in the manuscript:

“The sample size was estimated using a target power of 80%, at alpha of 0.05, and was calculated relative to the small effect size in the latent growth analysis using an a-priori sample size calculator [26]. A minimum of 125 patients should be sufficient to obtain the adequate power analyses. Considering the potential attrition rate of 20%, the final sample is 150 patients. Similar values (for two-level model with random slopes and within-person mediation model) come from simulation studies using the Monte Carlo approach [24,27].” (p. 10)

It is worth mentioning that 17 parameters will not be analyzed simultaneously. From our experience, even in a very large group (in this case it would have to be several thousand), with such a large number of variables, there would be a problem with model convergence. Our research hypotheses are testing the meaning-making model in pieces, taking into account key mediations and moderations. A group of about 150 people should be enough. If necessary, Bayesian estimation (standard on DSEM) will be used, which is recommended for small sample sizes. In studies in a clinical group, gathering a group of several thousand respondents is very difficult.

References:

Bolger, N., & Laurenceau, J.-P. (2013). Intensive longitudinal methods: An introduction to diary and experience sampling research. Guilford Press.

Mehl, M. R., & Conner, T. S. (2012). Handbook of research methods for studying daily life. Guilford Press.

Ochoa Arnedo, C., Sánchez, N., Sumalla, E. C., & Casellas-Grau, A. (2019). Stress and growth in cancer: Mechanisms and psychotherapeutic interventions to facilitate a constructive balance. Frontiers in Psychology, 10. https://doi.org/10.3389/fpsyg.2019.00177

Park, C. L. (2011). Meaning, coping, and health and well-being. In S. Folkman (Ed.), The Oxford handbook of stress, health, and coping (pp. 227–241). Oxford University Press.

Sagan, A. (2019). Sample size in multilevel structural equation modeling – the Monte Carlo approach. Econometrics, 23, 63–79. https://doi.org/10.15611/eada.2019.4.05

Shrout, P. E., & Lane, S. P. (2012). Psychometrics. In M. R. Mehl & T. S. Conner (Eds.), Handbook of research methods for studying daily life (pp. 302–320). The Guilford Press.

---

## [Decision Letter · Decision Letter 1]

28 Sep 2022

The role of psychological flexibility in the meaning-reconstruction process in cancer: the intensive longitudinal study protocol

PONE-D-21-36245R1

Dear Dr. Kroemeke,

We’re pleased to inform you that your manuscript has been judged scientifically suitable for publication and will be formally accepted for publication once it meets all outstanding technical requirements.

Kind regards,

Jamie Males

Editorial Office

PLOS ONE

Additional Editor Comments (optional):

Reviewers' comments:

Reviewer's Responses to Questions

**Comments to the Author**

1. Does the manuscript provide a valid rationale for the proposed study, with clearly identified and justified research questions?

Reviewer #1: Yes

2. Is the protocol technically sound and planned in a manner that will lead to a meaningful outcome and allow testing the stated hypotheses?

Reviewer #1: Yes

3. Is the methodology feasible and described in sufficient detail to allow the work to be replicable?

Reviewer #1: Yes

4. Have the authors described where all data underlying the findings will be made available when the study is complete?

Reviewer #1: Yes

5. Is the manuscript presented in an intelligible fashion and written in standard English?

Reviewer #1: Yes

6. Review Comments to the Author

You may also provide optional suggestions and comments to authors that they might find helpful in planning their study.

Reviewer #1: The modifications submitted are satisfactory. I wish the authors good success in the conduct of their study.

7. PLOS authors have the option to publish the peer review history of their article (what does this mean?). If published, this will include your full peer review and any attached files.

Reviewer #1: No

---

## [Editor Report · Acceptance letter]

4 Oct 2022

PONE-D-21-36245R1 

The role of psychological flexibility in the meaning-reconstruction process in cancer: the intensive longitudinal study protocol 

Dear Dr. Kroemeke:

I'm pleased to inform you that your manuscript has been deemed suitable for publication in PLOS ONE. Congratulations! Your manuscript is now with our production department. 

Kind regards, 

on behalf of

Dr. PLOS Manuscript Reassignment 

Staff Editor

PLOS ONE